# Creation of a Whole Health Age-Friendly Template and Dashboard Facilitates Implementation of 4Ms into Primary Care

**DOI:** 10.3390/geriatrics7050109

**Published:** 2022-10-01

**Authors:** James S. Powers, Natalie Penaranda

**Affiliations:** 1Geriatric Research, Education, and Clinical Center, Tennessee Valley Healthcare System, Nashville, TN 37212, USA; 2Division of Geriatrics, Vanderbilt University Medical Center, Nashville, TN 37212, USA; 3McGuire VA Hospital Central Virginia VA Health System (CVCHS), Geriatrics and Extended Care Service, Richmond, VA 23249, USA

**Keywords:** age-friendly care, geriatrics, electronic medical record, automated clinical dashboard

## Abstract

The Veterans Administration has joined the Age-Friendly Health Systems (AFHS) movement as part of its Whole Health initiative to provide safe, high-quality geriatric care using a set of evidence-based practices known as the “4Ms”—What Matters, Medication, Mentation, and Mobility—to provide care across all care settings. Two healthcare centers utilized an automated review of 4Ms care. For non-templated notes in the TVHS GeriPACT clinic over a 30-day period, all the 4Ms health factors (HFs) were addressed in only 1% of patients, and 16% had three HFs, 37% had two HFs, and 71% had one HF addressed. During the pilot of a new templated note and associated dashboard at the RICVAMC, GeriPACT and Home-Based Primary Care (HBPC) addressed all the age-friendly health factors in 41% of patients, while 24% had three health factors, 10% had two health factors, and 13% had one health factor addressed, and 10% were indeterminate by manual review. For both facilities, What Matters Most had the lowest prevalence, representing the most difficult individual health factor to address. The use of a templated note improves the reliable delivery of age-friendly care compared to non-templated notes and facilitates the dashboard display of practice- and provider-specific age-friendly encounter data, which may provide useful QI information to clinicians and health systems.

## 1. Introduction

As a High-Reliability Organization, the Veterans Administration has joined the Institute for Healthcare Improvement’s (IHI) Age-Friendly Health Systems (AFHS) movement as part of its Whole Health initiative to provide every older veteran with safe, high-quality care aligned with what matters most. Becoming an Age-Friendly Health System means that hospitals and healthcare systems reliably use a set of evidence-based practices known as the “4Ms”—What Matters, Medication, Mentation, and Mobility—to provide care for older, disabled, and medically complex patients across all care settings. When provided together in clinical practice, the 4Ms represent a broad cultural shift toward patient-centered care consistent with the mission of focusing on what matters most to individuals [1].

Implementing the IHI 4Ms age-friendly principles into primary care is challenging because there is no best practice to identify the documentation of delivery or to provide feedback to providers. Leveraging the electronic health record (EHR) is a potential way to automate this process. Note templates facilitate the incorporation of information into the EHR, whereas dashboards represent information management tools which use data visualization to display performance indicators to facilitate the tracking of performance [2]. We describe our experience at two VA medical centers, the Tennessee Valley Healthcare System (TVHS) and the McGuire Richmond VAMC (RICVAMC), utilizing both standard notes and templated notes and an operational dashboard to display 4Ms quality metrics.

## 2. Materials and Methods

The TVHS does not have a templated note or age-friendly dashboard. To automate the assessment of 4Ms care, age-friendly principles for primary care were discussed at IHI peer coaching webinars, and PDSA cycles were employed among clinic staff to define each of the 4Ms for the TVHS geriatric outpatient practice’s Geriatric Patient-Aligned Care Team (GeriPACT). SQL scripts were used to build field and note title searches (i.e., clinical reminders and medication reconciliation notes) for the 4Ms, defined as: (1) Mentation—mini-COG and PHQ2 (nursing intake), (2) Medication—medication review and provider reconciliation, (3) Mobility—ADL mobility questions (nursing intake), and (4) What Matters Most—patient portal messages, as the utilization of a patient portal in an integrated health system relates to functional engagement with the patient. Clinic visits in April 2021 were assessed for 4Ms age-friendly care in the TVHS GeriPACT. Plan-Do-Study-Act (PDSA) cycles with stakeholder clinicians reviewing dashboard-derived data refined the searches with improvement in accuracy. Templated notes to document 4Ms care were not utilized.

The RICVAMC uses a local templated 4Ms documentation tool designed to abstract data for QI and to standardize care across settings, which is incorporated into local note templates. Simultaneously, 4Ms health factors were developed by the VA Office of Patient Care and Cultural Transformation, and added to the Whole Health—Health Factors Dashboard (WH4ALL) [3] which is based on VA HSRD Standardized Health Factor Data Sets from the VA Central Data Warehouse (CDW) [4]. Age-friendly health factors (Table 1) are addressed by clinicians during patient care, radio-button flagged during the encounter documentation, and pulled into CDW. Some or all age-friendly health factors may be addressed during the encounter. An SQL data-base relationship cube (Pyramid Analytics) with weekly updates from the CDW was developed based on these identified fields. A Power Business Intelligence (Power BI) interface with multi-select dropdown filters was utilized to develop a dashboard and to permit searches including facility, provider, and clinic visit stop codes. (Figure 1) Clinicians utilizing the EHR were involved in every stage of the development of the dashboard, providing usability recommendations as well as direct feedback to the development team.

## 3. Results

The dashboard displays for the age-friendly WH4ALL and the RICVAMC-specific dashboards are provided in Figure 1 and Figure 2.

For non-templated TVHS GeriPACT clinic notes over a 30-day period, all the 4Ms health factors (HFs) were addressed in only 1% of patients, and 16% had three HFs, 37% had two HFs, and 71% had one HF addressed. During the pilot of a new templated note at the RICVAMC, GeriPACT and Home-Based Primary Care (HBPC) addressed all the age-friendly health factors in 41% of patients, while 24% had three health factors, 10% had two health factors, and 13% had one health factor addressed, and 10% were indeterminate by manual review. (Table 2) For both facilities, What Matters Most had the lowest prevalence, representing the most difficult individual health factor to address.

## 4. Discussion

Leveraging the EHR to display the simultaneous documentation of 4Ms for a primary care population could facilitate improved provider-driven interventions to provide 4Ms care to older adults. The Agency for Healthcare Research and Quality (AHRQ) utilizes the Common Elements for Event Reporting-hospitals (CFER-H) in developing the National Patient Safety Data Dashboard [2]. A dashboard derived from data generated from templated notes could be a powerful tool to inform clinical activity and help busy clinicians promote age-friendly patient-centered care and Patient Priorities Care, which helps patients and clinicians focus all their decision-making and healthcare on what matters most: patients’ own health priorities [5]. Dashboards may also display important clinical parameters that are helpful for documenting the association between 4Ms care and other important clinical outcomes.

Local clinical access coordinators can add the standardized 4Ms template containing age-friendly health factors to any EHR progress note template or as a text reminder by identifying the desired health factor component and creating a text integration utility (TIU) using the selected health factor object(s) [6]. Clinicians are able to use their own customized template to address each health factor or can incorporate the standardized 4Ms template. 

Local SQL script searches are labor-intensive and not generalizable. Dashboards connected to the CDW permit the standardization and dissemination of data. Age-friendly note templates can be developed that are appropriate for all levels of care, including primary care, inpatient, long-term care (Community Living Centers—CLCs), home care (Home-Based Primary Care—HBPC) and consultation use. A dashboard display derived from encounter flags permits clinicians to customize their delivery of 4Ms care and clinical documentation and may be superior to pop-up alerts, which subject clinicians to alert fatigue. Dashboards may also be useful in identifying older or high-risk patients in a general population to facilitate targeting 4Ms care.

Building, standardizing, and disseminating age-friendly dashboards and data capture could help health systems to incorporate 4Ms care, identify barriers and target populations for implementation, and inform best practice to enhance the delivery of age-friendly care to the older adult.

## 5. Limitations

Work continues to educate providers about incorporating 4Ms templates into their clinical documentation to facilitate age-friendly workload credit and facility QI monitoring. The 4Ms Health Factor enhanced WH4ALL dashboard does not search other fields such as note titles and clinical reminders and does not include a natural language-processing (NLP) text-search function to identify elements of age-friendly care within clinical notes.

## 6. Conclusions

An operational dashboard may help inform the delivery of 4Ms care. The use of a templated note improves the reliable delivery of age-friendly care compared to non-templated notes. A dashboard display of practice- and provider-specific age-friendly encounter data could provide useful QI information to other clinicians and health systems. Patient portals may be a potential source of information regarding patient–clinician engagement.

## Figures and Tables

**Figure 1 geriatrics-07-00109-f001:**
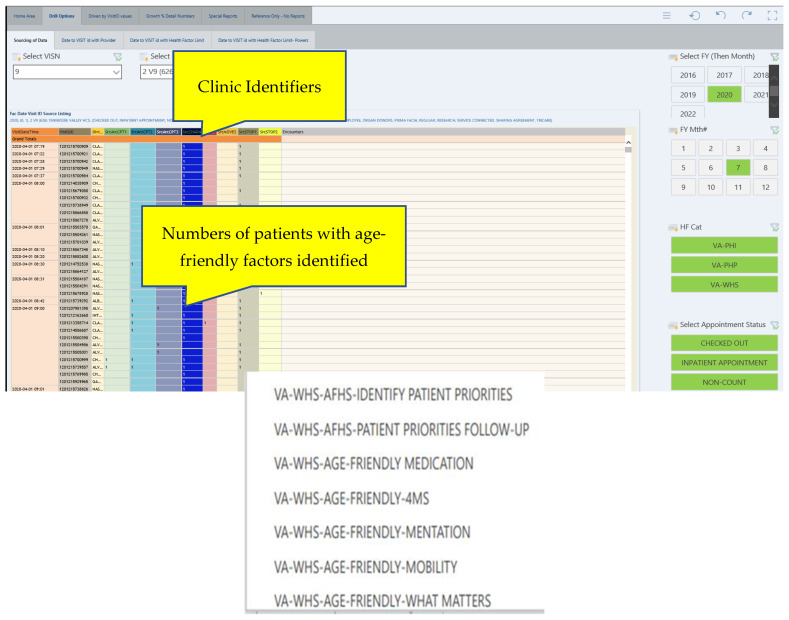
Whole Health age-friendly dashboard (WH4ALL). Display visualizes the opening page of the national age-friendly dashboard with health factors addressed in the electronic health record by locality (facility) for specified time periods, with drill-down capability for specific clinic population or provider.

**Figure 2 geriatrics-07-00109-f002:**
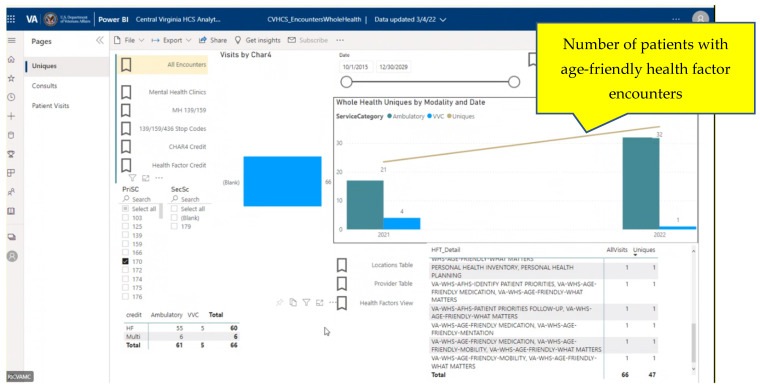
Whole health age-friendly dashboard (RICVAMC). Display visualizes the RICHVAMC age-friendly dashboard with health factors addressed over time in the Geriatric Primary Care Patient-Aligned Care Team (GeriPACT) population and Home-Based Primary Care (HBPC).

**Table 1 geriatrics-07-00109-t001:** Age-friendly whole health factors.

Health Factor (HF) Name	Definition	Application
VA-WHS Age-Friendly 4Ms	The 4Ms framework (What Matters, Medication, Mentation, and Mobility) is an essential set of evidence-based practices needed to provide age-friendly care to older veterans.	Applied when all the 4Ms are assessed and acted on in a single patient encounter.
VA-WHS Age-Friendly What Matters	“What Matters” is the first “M” for providing age-friendly care. What Matters is asked and documented, and the veteran’s care plan is aligned with their health outcome goals and care preferences.	Applied when What Matters is asked, documented, and acted on in a single patient encounter.
VA-WHS Age-Friendly Medication	“Medication” is the second “M” for providing age-friendly care. High-risk medications are reviewed, documented, deprescribed, and avoided. If Medication is necessary, the Medication does not interfere with What Matters to the older veteran, their Mobility, or Mentation.	Applied when Medication is assessed and acted on in a single patient encounter.
VA-WHS Age-Friendly Mentation	“Mentation” is the third “M” for providing age-friendly care. There is an awareness of cognitive impairment warning signs to prompt the evaluation of cognition. Depression and delirium are prevented, identified, treated, and managed. Changes in mood or mental health are identified and managed.	Applied when all elements of Mentation are assessed and acted on in a single patient encounter.
VA-WHS Age-Friendly Mobility	“Mobility” is the fourth “M” for providing age-friendly care. Mobility limitations are identified and it is ensured that each older veteran moves safely every day to maintain function and do What Matters.	Applied when Mobility is assessed and acted on in a single patient encounter.

**Table 2 geriatrics-07-00109-t002:** Addressing age-friendly health factors in outpatient care.

*n* = 127 TVHS GeriPACT Non-Templated Notes			
Health Factor Addressed	*n* (%)	Number of Health Factors Addressed	*n* (%)
What Matters	19 (15%)	4 HFs	1 (1%)
Mentation	36 (24%)	3 HFs	16 (13%)
Medication	64 (50%)	2 HFs	38 (30%)
Mobility	68 (54%)	1 HFs	72 (56%)
***n* = 49 RICVAMC GeriPACT and HBPC Templated Notes**			
**Health Factor Addressed**	***n* (%)**	**Number of Health Factors Addressed**	***n* (%)**
What Matters	11 (22%)	4 HFs	20 (41%)
Mentation	14 (29%)	3 HFs	12 (24%)
Medication	12 (24%)	2 HFs	5 (10%)
Mobility	12 (24%)	1 HFs	7 (13%)
Indeterminate	5 (10%)		

## Data Availability

Data is available from the corresponding author at james.powers@vumc.org.

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
