# Peer review of "Creation of a Whole Health Age-Friendly Template and Dashboard Facilitates Implementation of 4Ms into Primary Care"

_geriatrics, 2022, doi:10.3390/geriatrics7050109_

Round 1

Reviewer 1 Report

Thank you for this important work exploring how using standard notes, templated notes and an operational dashboard in the EHR can be used to incorporate information on the 4M's of age friendly care. Operationalization of the 4M's is important in order to improve care for older adults. 

The work that you have done at two VA medical centers is impressive. However, the manuscript describing this work would contribute more to the knowledge base regarding 4M's care if the following were addressed:

1. I recognize that there is limited space for a Communication article. However, including a reference to the VA initiatives around AF Health Systems would be appropriate. 

2. Providing some background on the 2 VA Medical Centers would be helpful. I assume these are VA hospitals that the GWEP is working with? What is the role of the GWEP in this work? 

3. Figure 1 is difficult to understand without more context. In addition, I am concerned the visit number and date could potentially identify patients. Is this Figure needed? Could it be portrayed in a way that provides more useful information to the reader - perhaps arrows and notes describing the fields? 

4. Why are the 4M's and What Matters rows a different color in Table 1. I feel like the definition of the M's as health factors could be placed in the text and save space. While it is important to understand the 4Ms, I am not sure what additional information the table is providing, particularly in regards to Application. 

5. Similar to my comments regarding Figure 1, Figure 2 is not intuitive to read. Providing more description of the dashboard (perhaps breaking it down) would be useful. 

6. Table 2 3rd column is not labeled. The health factors addressed N for each row is different than described in the body. 

7. I hoped to read a discussion about the advantage of dashboards compared to templated notes and how both can be used to provide information on age friendly successes. 

Author Response

Reviewer 1  author responses highlighted

  1. I recognize that there is limited space for a Communication article. However, including a reference to the VA initiatives around AF Health Systems would be appropriate.

The introduction has been edited for clarity and now indicates that the VA has adopted the Institute for Healthcare Improvement’s age friendly movement, with reference provided

  1. Providing some background on the 2 VA Medical Centers would be helpful. I assume these are VA hospitals that the GWEP is working with? What is the role of the GWEP in this work? 

Clarification is provided that TVHS has no templated note for age friendly dashboard while RICHVAMC develop both a templated note and associated age friendly dashboard. HRSA Grant funding permitted SQL programming to search indicators of 4Ms care in TVHS notes to provide a comparison to data obtained from RICHVAMC

  1. Figure 1 is difficult to understand without more context. In addition, I am concerned the visit number and date could potentially identify patients. Is this Figure needed? Could it be portrayed in a way that provides more useful information to the reader - perhaps arrows and notes describing the fields? 

Text boxes are added to clarify the view and data displayed in the figures. The legends are further edited for clarity. Access to PHI is not a concern as utilization of the drill down function to identify patients must be performed while utilizing the VA site by registered users.

  1. Why are the 4M's and What Matters rows a different color in Table 1. I feel like the definition of the M's as health factors could be placed in the text and save space. While it is important to understand the 4Ms, I am not sure what additional information the table is providing, particularly in regards to application. 

Table 1 is edited and color removed. This table was selected as a concise method to demonstrate each of the age friendly health factors.

  1. Similar to my comments regarding Figure 1, Figure 2 is not intuitive to read. Providing more description of the dashboard (perhaps breaking it down) would be useful. 

A text box is added to figure 2 and the legend edited to help clarify the data displayed

  1. Table 2 3rd column is not labeled. The health factors addressed N for each row is different than described in the body. 

Table 2 is reformatted to improve clarity

  1. I hoped to read a discussion about the advantage of dashboards compared to templated notes and how both can be used to provide information on age friendly successes. 

The discussion is edited to clarify that the dashboard is derived from the templated note in order to provide direct feedback to providers.

Reviewer 2 Report

The topic has relevance and is important to improve overall care for veterans. The idea of conducted research is important to better introduce 4M concept in the primary care process. Conducted research for sure can help to push using of 4M concept.

The overall article design requires improvements to be acceptable for publication and to make interest for reading in the community.

The introduction must be better structured to better explain purpose of the article and later conducted research.

The purpose of figure 1 is not clear - visibility of the portal is bad in the printed version - in the digital version with big magnify is still hard to recognize the real purpose of the provided data on the portal - in general is missing description of the portal interface.

Analysis of achieved results on the conducted research are important but presentation of the achieved results must be improved based on the achieved data.

It is very important to have such reports and improvement in the presentation of the conducted research and results presentation needs better polishing in presentation.

Author Response

Reviewer 2 author responses highlighted

The introduction must be better structured to better explain purpose of the article and later conducted research.

The introduction and materials and methods sections have been edited to enhance clarity. RICHVA utilized both a template and associated dashboard.

The purpose of figure 1 is not clear - visibility of the portal is bad in the printed version - in the digital version with big magnify is still hard to recognize the real purpose of the provided data on the portal - in general is missing description of the portal interface.

Text boxes are added to clarify the view and data displayed in the figures

Analysis of achieved results on the conducted research are important but presentation of the achieved results must be improved based on the achieved data.

The discussion has been edited for clarity

It is very important to have such reports and improvement in the presentation of the conducted research and results presentation needs better polishing in presentation.

The abstract and conclusion have been edited for clarity

Round 2

Reviewer 1 Report

Thank you for your concise responses to editorial comments. I look forward to this article being in press as it will be important for other VAs to reference as they embark on the Age Friendly Health Systems journey.